# Strategies for Natural Products Discovery from Uncultured Microorganisms

**DOI:** 10.3390/molecules26102977

**Published:** 2021-05-17

**Authors:** Khorshed Alam, Muhammad Nazeer Abbasi, Jinfang Hao, Youming Zhang, Aiying Li

**Affiliations:** Helmholtz International Lab for Anti-Infectives, Shandong University-Helmholtz Institute of Biotechnology, State Key Laboratory of Microbial Technology, Shandong University, Qingdao 266237, China; microkhorshed@mail.sdu.edu.cn (K.A.); nazeer.abbasi1990@gmail.com (M.N.A.); h15935152510@163.com (J.H.)

**Keywords:** natural products, uncultured microorganisms, metagenomics, extreme ecosystem, heterologous expression, eDNA, microbial dark matter, environmental metabolome, single cell sequencing

## Abstract

Microorganisms are highly regarded as a prominent source of natural products that have significant importance in many fields such as medicine, farming, environmental safety, and material production. Due to this, only tiny amounts of microorganisms can be cultivated under standard laboratory conditions, and the bulk of microorganisms in the ecosystems are still unidentified, which restricts our knowledge of uncultured microbial metabolism. However, they could hypothetically provide a large collection of innovative natural products. Culture-independent metagenomics study has the ability to address core questions in the potential of NP production by cloning and analysis of microbial DNA derived directly from environmental samples. Latest advancements in next generation sequencing and genetic engineering tools for genome assembly have broadened the scope of metagenomics to offer perspectives into the life of uncultured microorganisms. In this review, we cover the methods of metagenomic library construction, and heterologous expression for the exploration and development of the environmental metabolome and focus on the function-based metagenomics, sequencing-based metagenomics, and single-cell metagenomics of uncultured microorganisms.

## 1. Introduction

### 1.1. Natural Products and Uncultured/Uncharacterized Microbes

Microbial natural products (NPs) are specialized metabolites, commonly treated as unprecedented important sources and perform a prominent impact, especially in pharmaceutical research and production [1,2,3]. Systematic exploration and biosynthetic elucidation of NPs is therefore an important turn against studying complex biodiversity and leveraging those chemical varieties for use in medicine, food, farming, and environmental safety [4,5]. However, due to the rediscovery of NPs in high rate by conventional methods, microorganisms ever seemed to have been depleted sources for following NP-mining for almost a century [6]. The latest boom of microbial genome sequences indicates a huge unexplored treasure trove of novel NPs [7]. Many new NPs have been uncovered through the mining of microbial genetic data by following recently developed synthetic biology-inspired strategies and tools. However, the genomic period has shown that cultivated bacteria constitute only a small portion of the overall known biomass of bacteria.

Microorganisms can inhabit as free living organisms or symbiotic associations with multicellular hosts or other unicellular organisms in various ecological habitats, from deepest seas to arid deserts and the guts of the multicellular organisms, even some symbiotic or parasitic systems [8]. It has been estimated that 4 × 10^6^ different microbial taxa and 10^9^ cells are present per ton and per gram of soil samples, while microbial soil diversity is estimated to have between 3000 and 11,000 genomes per gram of the soils [9]. However, less than 1% of the microbes of the ecosystems can be accessible through cultivation techniques and more than 99 % cannot be studied, restricting our knowledge of microbial morphology, metabolism, genetics, and population ecology of uncultured microorganisms [10]. Maintaining or imitating natural growth conditions for these uncultured microorganisms is still a challenge [9]. Thus, it has been a challenge to identify the NPs from those uncultured or uncharacterized microbes for a long time.

### 1.2. Culture-Independent Tools for the Study of Uncultured Microorganisms Related to NP Accumulation

The advent of next-generation sequencing technology and comprehensive use of metagenomics-related techniques has made uncultured microorganisms accessible for functional research: (i) function-based metagenomics based on library construction, heterologous expression and functional screening, (ii) sequencing-based metagenomics [11], and (iii) single-cell metagenomics [12] offer alternate, culture-independent approaches, allowing researchers to discover complex microbial flora, novel taxa, and explore dark or hidden NPs from these uncultured microorganisms [13,14]. Insights into unusual enzymology in the context of NP biosynthesis suggested that the huge diversity of NPs awaits discovery [15]. Combining synthetic biological approaches with these metagenomics techniques will promote their wider application into the exploration of NPs from these uncultured microorganisms [14].

#### 1.2.1. Evaluation of Microbial Heterogeneity and Abundance in Environmental Samples

To evaluate the heterogeneity and abundance of microbes with potentials to produce NPs in environmental samples, 16S rRNA gene sequencing has been used to classify uncultured microbial diversity. Moreover, some other functional genes have demonstrated higher accuracy for the analysis of genetic variation in the normal bacterial populations including genes *soxB* and *amoA* as unique genes to sulfur oxidizing bacteria and ammonia oxidizing microbes, and genes encoding conserved domains in polyketide synthetases (PKS) or nonribosomal peptide synthetases (NRPS) for NP-producing bacteria, and so on [12,16,17,18,19,20].

Because metagenomics avoids the isolation and propagation of organisms and does not entail prior understanding of the organisms present in the samples, its application facilitates researchers to look at the broad range of individual genes and whole operons for biosynthetic or degradative pathways, and thus promotes the biochemical characterization of diverse natural samples and the discovery of unique bioactive compounds and enzymes from environmental DNA samples [21]. In the first microbial metagenomics project, Venter and colleagues reported 1800 genomic organisms, comprising 148 new species phylotypes within water bodies of the Sargasso Sea [22].

#### 1.2.2. Metagenomics-Related Approaches

Since 1998, metagenomics has appeared as a concept and been developed into important tools for gaining insight into the morphology and genetics of uncultured species [21].

For function-based metagenomics, the construction of a metagenomic library in a suitable host is the first step by cloning fragmented environmental DNA (eDNA) into expression systems and then introducing these metagenomic clones into a heterologous host [23]. Next, screening targets of interest genes or gene clusters could be conducted using function-based screening. Biosynthetic pathways of NPs of uncultured microorganisms can be obtained for the detection of compounds through the design and deployment of a metagenomic library, metabolic screens, and color screens (Figure 1).

For sequencing-based metagenomics, total eDNA isolation and sequencing, pathway predication and designing, and DNA assembly are required, which follow heterologous expression of target clones (Figure 1).

Many excellent reviews on function-based metagenomics [24], sequencing-based metagenomics [25], and their applications in biotechnology are available. In particular, the number of a particular species in the metagenomic library is determined by factors such as genome size, genome copy number, within-species heterogeneity, and relative frequency of the species as well as biases in DNA extraction and sequencing.

Single-cell metagenomics is also an effective tool for insights into the taxonomy, physiology, and biochemistry of specific uncultured species in environmental samples [12,26,27,28] (Figure 1). It depends on the sorting of cells present in environmental samples, single-cell genomic DNA isolation, amplification and sequencing, and target BGC assembly and expression. Recently, it has been used for the exploration of NPs from different environmental samples such as marine animal tissues.

## 2. Function-Based Metagenomics for Exploration of NPs from Different Environmental Samples

This metagenomic analysis focused on the cloning of eDNA into expression vectors and their propagation in suitable hosts, accompanied by an evaluation of recombinant action. Metagenomic gene libraries may be used to identify novel enzymes encoded by a single gene or a small operon, whereas large insert libraries are necessary for the extraction of large BGCs that encode complex pathways containing multiple genes for the production of NPs [29].

### 2.1. Sample Preparation and Evaluation, Library Construction, and Heterologous Expression

Generally, it is necessary to enrich and evaluate environmental samples prior to the creation of the metagenomic library to improve the performance of gene screening through metagenomic analysis (Figure 1). Aided by next generation sequencing, PCR amplification of particular functional genes using degenerated primers was used to study the genetic variation in the environmental samples while amplification of 16S rRNA by PCR was applied to evaluate the genetic variety in prospective environmental samples for the development of metagenomic libraries.

Additionally, a microarray technology was also used to evaluate and disseminate functional genes through DNA array technology. Suppressive subtractive hybridization (SSH) was used to assess microbial heterogeneity and functional discrepancies in genetic information among different environments. Stable isotope sampling probing (SIP) was used to identify DNA and RNAs of each gene [30]. After that, the tagged DNA and RNAs can be isolated using density gradient centrifugation. Multiple displacement amplification (MDA) has been applied to improve the environmental yield of DNA from low-biomass environmental samples [23].

In order to increase the possibility of extracting NP complete gene clusters during the construction of metagenomic libraries, the library size and gene cluster structure need to be considered carefully before library construction. Thus, methods divided as direct or indirect extraction procedures for separating DNA from environmental samples like soils/sediments have been developed [31]. Direct DNA isolation methods depend on the in situ lysis of microbial cells in the specimens before retrieval and extraction of DNA. Indirect DNA isolation techniques have been established to shield DNA from mechanical stress, which can cause DNA trimming and allow large fragments to be obtained.

Short insert libraries will not promote the identification of larger gene clusters (20–200 kb) or operons for NPs. To address this restriction, bacterial artificial chromosome (BAC) vectors have been acclaimed for creating large-insert libraries (up to 200 kb).

*Escherichia coli* strain is a better host for the cloning and expression of Gram-negative bacterial enzymes, but not perfect for the cloning and expression of Gram-positive bacterial BGCs with high GC content. In order to improve the biosynthetic variability and expression-related problems defined in function-based metagenomic studies, a variety of study teams have established expression mechanisms focused on non-*E. coli* hosts.

The Brady group evaluated the capacity to produce similar biosynthetic genes with pigmented or antibacterial phenotypes for a variety of host strains [32,33]. In most instances, only the hosts where genes were initially extracted or close-phylogenetically hosts were recorded to show activity or functional expression of foreign genes. This finding highlights the important role of developing alternate hosts in the exploration of NPs powered by metagenomics. *Streptomyces lividans*
*(S. lividans)*, *S. albus*, *Pseudomonas putida*
*(P. putida**)*, *Ralstonia metallidurans*, *Agrobacterium tumefaciens*, *Burkholderia graminis*, and *Caulobacter vibrioides* bacteria have been used as eDNA recipient hosts [34].

To increase heterologous expression of target genes or BGCs located in the metagenomic libraries, some regulators were introduced into heterologous hosts. For example, introduction of heterologous sigma factors could significantly improve expression of eDNA [35]. The *trp* operon and genes employed in the biosynthesis of new antibiotics were identified using *S. lividans*, *R. leguminosarum*, and *P. putida*. *R. leguminosarum* was used as a suitable host for the expression of functional genes. *S. lividans* was often used for the expression of DNA derived from the soil. Five novel molecules, terragins A–E, along with siderophoric nocardamine, have been evaluated for these strains. *P. putida* is not a suitable host for screening the inhibition zone [32].

Advanced BAC shuttle vectors were constructed for high-throughput conjugation of big threads of eDNA from *E. coli* to *S. lividans* and *P. putida*. Martinez observed that genes responsible for the production of various antibiotics in *E. coli*, *P. putida*, and *S. lividans* were expressed differently using this shuttle vector [36]. A variety of polyketides were obtained from *R. metallidurans*, which were new [37] and two were amides gained by antibacterial screening [32].

### 2.2. Function-Based Screening

Function-based screening in a metagenomic study is a potent, but challenging technique that requires the creation of an expression library first, and next the measurement of a particular reaction with a substrate to evaluate the biochemical and metabolic functions of relevance [38]. A variety of antibiotics (Table 1), hydrolytic enzymes, antibiotic-resistance genes, Na(Li)/H transporters, and several other functions have been successfully identified using function-based metagenomics [23]. Furthermore, it permitted the identification of genes with specific behavior encoding enzymes, which are entirely new sequence forms with no known counterparts. Some extreme ecosystems have been chosen for study and allow for the detection of enzymes with the expected properties [39]. For instance, it might be predicted that thermal spring metagenome will contain unique genes that encode thermostable enzymes. The method’s benefit is that it needs no sequence analysis of the genes of interest, rendering it capable of identifying completely new groups of genes for unknown or unknown roles.

#### 2.2.1. Approaches for Function-Based Screening

Function-based screening is capable of the identification of new gene groups that encode new BGCs for natural compounds and functional enzymes. The identification of clones with changed phenotypes is one of the simplest screenings. These optical displays may be focused on pigmentation (color), development of inhibition zones around clones of metagenomic library forming on bacterial or fungal test lawns or blood agar plates as a result of biocatalytic conversion (production of antimicrobial or hemolytic factors) [34]. Along with turbomycin A (**1**, Figure 2) originally identified from a fungus, the red broad-spectrum antibiotic turbomycin B (**2**, Figure 2) [43] was extracted, as an example of a new NP uncovered by color/hemolytic screening. *N*-acyl amino acid derivatives and structurally related drugs as well as palmitoyl putrescine were among the first NPs identified using inhibition zone-based screening [44,45,46]. However, initially, metagenomic libraries had a low sensitivity and low performance based on function-based screening.

The advent of hybrid practical “intracellular” screening methodologies to distinguish clones of concern like METREX (metabolite-regulated expression) [47], SIGEX (substrate- induced gene expression) [48], and PIGEX (product-induced gene expression) [49] has sped up the extraction of new biocatalysts from microbial communities. Fluorescence-activated cell sorting (FACS) or fluorescence microscopy was used for these screening methods. FACS has a wide variety of uses in high-throughput metagenomic clone screening since it can detect biological activity within a single cell [50].

METREX, a framework wherein metagenomic DNA is embedded with a biosensor in a host cell, which can trigger bacterial quorum sensing [47,51]. The METREX system was used to retrieve quorum-sensing inducer-generating metagenomic clones from soil and activated sludge metagenomic libraries [38]. Green fluorescent protein (GFP) expressed by the metagenomic clone in the host cell can be observed by using fluorescent microscopy or a spectrophotometer. This technique succeeded in the preliminary detection of the recognized *N*-(3-oxohexanoyl)-l-homoserine lactone (**3**, Figure 2) and an unknown indoxyl degradation agent developed by a DNA library from soil and gypsy moth, *Lymantria dispar* [47,52].

SIGEX is another intracellular method of screening, based upon the understanding that catabolic gene expression is normally triggered by appropriate substrates and, in many cases, regulated by regulatory elements. A shotgun cloning with an operon-trap gfp-expression vector was developed to allow the collection of positive clones in liquid cultures using fluorescence-activated cell sorting. The cloning of new aromatic hydrocarbon-induced genes from a groundwater metagenomic library was illustrated using this screening platform [53].

Some studies have reported lately on high efficient screening of novel enzymes and natural compounds from a metagenomic library using microfluidic technique, despite the fact that the traditional isolation of active clones was done using an in vitro method [54,55,56,57]. Hosokawa et al. unveiled individual cells embedded in gel microdroplets (GMDs) for the screening of lipolytic enzyme genes from the soil-based metagenomic library [58]. Such findings showed the capacity of microfluid systems as an effective method for detecting new metagenomic enzymes, and minimizing time and cost.

In 2011, Jiang et al. noticed a new β-glucosidase gene (bgl1D) with lipolytic activity (renamed Lip1C) that was detected by a function-based screening of a soil-based metagenomic library [59]. Lipase and esterase continue to be the most specific enzyme activities using the function-based screening of diverse metagenomic libraries [60,61]. Some of the early metabolites showed intriguing changes that lead to enamine and enolether moieties. Isocyanide (**4**, Figure 2) is another antibiotic with a unique structure [62].

Chemical analysis such as high-performance liquid chromatography-mass spectrometry (HPLC-MS), is an effective yet time-consuming version of a function-based screening for identifying compounds in samples [42,63]. The screening time can be reduced by first assessing pools of clones. Biosynthetic genes for patellamide D (**6**, Figure 2) and ascidiacyclamide (**5**, Figure 2) have been discovered using a BAC library developed from the DNA of uncultured tunicate-derived cyanobacterial symbionts [63].

Although the results of its phenotypic screening of *E. coli*-based libraries have been promising in terms of innovative enzymology, the rate of positive clones and the structural variability of the molecules identified are usually limited. Brady et al. observed antimicrobial properties in one out of every 10,000 to 20,000 eDNA cosmid clones that express antibiotic proteins and ribosomal peptides [45]. An antifungal screening using *S. cerevisiae* revealed only one successful clone out of over 110,000 fosmid clones [64]. The understanding that heterologous expression of eDNA can lead to the exploration of new fascinating biochemistry was among the most intriguing aspects of these studies.

#### 2.2.2. Limitations of Function-Based Screening

Function-based screening is a simple system that allows researchers to access an immense genetic resource in a microbial population through the expression of cloned genes from metagenomic origin in heterologous hosts, without knowledge of the origin of the microbe or original gene sequence, or NP structure, or composition of the target proteins. However, the lack of direct and powerful detection methods for certain products and a limited selection of hosts for the expression of certain foreign genes or gene clusters have been major drawbacks in function-based screening.

First, the variety of functional activities is restricted to current screening methods like antibiotics or hydrolytic enzyme identification. Second, it is hard to find a suitable host able to post-translationally alter apo proteins and heterologously express all genes of one BGC. In particular, those genes derived from evolutionarily distant uncultured species cannot be effectively expressed in standard hosts as *E. coli*, due to divergent usage of codon, incompatible regulatory factors, lack of biosynthetical precursors, and a drawback to the extraction of full information from function-based screening [38]. Recently, new transformation systems using various microbes with additional platforms of gene expression and a variety of protein secretion mechanisms have been reported. Next, a constraint in size for metagenomic libraries was introduced. A fosmid library of 50,000 clones with an average insert size of 40 kbp equals around 500 bacterial genomes (4 Mbp size), even while 1 g of soils could constitute thousands of varying microbial species [65].

The carrier protein (CP) domains of multimodular nonribosomal peptide synthetases (NRPSs) and polyketide synthases (PKSs) enzymes act as anchors to the developing natural product chain. In standard *E. coli* expression hosts, CPs are unresponsive until they are translated into holo domains by attaching a 4-phosphopantetheinyl unit [66]. Moreover, *E. coli* lacks the ability to synthesize methylmalonyl-CoA (coenzyme A), a necessary component for the development of several methyl-branched polyketides, like antibiotic erythromycin [67]. *E. coli* can be transformed into an effective producer of complex PKSs and NRPSs by the incorporation of genes for Phosphopantetheinyl transferases (PPTs) and for branched PKSs methylmalonyl-CoA [67]. However, due to the large size of the most complex PKSs and NRPSs gene loci, which outstrips the normal fosmid and sometimes even BAC inserts, no efficient functional exploration methods for these compounds are known to date.

## 3. Sequencing-Based Metagenomics

In contrast to function-based metagenomics, sequencing-based metagenomics does not need heterologous expression to distinguish metagenomic clones of consideration. Additionally, even if NP pathways are not observed in the library host, they can be identified. This method avoids the difficulties of heterologous expression by using sequence analysis to classify gene clusters of interest. After that, heterologous expression experiments can be used to produce molecules based on the target pathways. Some examples of NPs screened by such a sequencing-based approaches were identified from symbiotic systems (Table 2, Figure 3), where most of the microorganisms cannot live freely, but in a host-dependent mode.

Early implementations of sequencing-based approaches employed primers specific to genes preserved in comparatively limited NP biosynthesis families. Several of the latest studies have been carried out to increase such endeavors in order to identify a wider variety of retained biosynthetic NP domains with an important emphasis on domains found among biosynthetically modified PKS and NRPS genes [68,69,70,71]. For example, for large multimodular PKS and NRPS clusters that typically surpass the size of fosmid or even BAC inserts, a sequencing-based screening is the best suitable approach.

Another important example is the discovery of cadasides (CDE) from a soil sample: the Brady SF group used NRPS adenylation (AD) domain sequencing to guide the identification, recovery, and cloning of the *cde* BGC, which led to the production of cadasides A and B (13, Figure 3), a subfamily of acidic lipopeptides. They found that sequencing of AD domains from diverse soils revealed that these sequences predicted to arise from cadaside-like gene clusters are predominantly found in soils containing high levels of calcium carbonate [72] (Table 2).

Sequencing analysis driven by the discovery of phylogenetic markers is a powerful technique first introduced by the DeLong group, which provided the first genomic sequence related to the 16S rRNA gene of an uncultured archaeon [73]. Targeted gene sequencing or shotgun metagenome sequencing are two commonly used platforms in microbial metagenomics [74]. When targeting genomics, eDNA is extracted and purified using highly effective DNA isolation and purification methods. The selected genes are multiplied with oligonucleotide markers and sequencing adaptors, specifically designed sequences for the combined various specimens [75]. In a shotgun sequencing method, the metagenomic DNA is segmented, finally reconstructed, and connected to adapters that enable template replication and eventual sequencing to produce a large number of short readings that can be subsequently constructed in silico, guiding to the creation of NP BGCs of interest using different computational tools and synthetic biology methods [75].

**Table 2 molecules-26-02977-t002:** Natural products identified/studied by sequencing-based approaches.

Compound	Sources	Mode of Action	Reference
Bryostatin ^#^	*Bugula neritina*	Antitumor	[76]
Pederin ^#^	*Paederus fuscipes*	Antibacterial	[77,78]
Onnamide ^#^	*Theonella swinhoei*	Antitumor	[79]
Cadasides	Soil sample	Antibacterial	[72]

Note: ^#^ indicating these compounds were isolated via sequencing-based screening from uncultivated microorganisms that live in symbiotic systems.

Degenerate primers that target certain typical biosynthesis domains are constructed to create intricate PCR amplicons from gene clusters contained in specific microbial communities or metagenomic libraries. Specific next generation sequencing (NGS) readings obtained from such PCR amplicons are referred to as natural product sequence tags (NPSTs) [11]. The data provided by these studies have resulted in new bioinformatic technologies, which can affirm and arrange NPST databases. Although a range of bioinformatics tools have already been designed to mine full sequenced genomes for secondary metabolite biosynthetic clusters, these programs typically need much greater DNA fragment inputs than metagenomic sequencing efforts provide.

Programs like eSNaPD (Environmental Surveyor of Natural Product Diversity) [71] and NaPDoS (Natural Product Domain Search) [80] were designed to evaluate very large NPST datasets. Such bioinformatics tools interpret NPSTs with standard libraries of sequences collected from the BGCs to determine the output of the gene clusters and the possible chemical structures of the environmental samples or libraries. This method is related to the reconstruction of whole species within the ecosystem using 16S rRNA sequences [81].

Two routes for environmental NP mining are established by the phylogenetic assembly of BGCs (Figure 1):

(i) Constituents of characterized NPs may be mined through the pursuit of NPST-associated gene clusters that are closely linked to biosynthetic domains that encode recognized natural products [72,82,83,84]. Comparative searches of these near and distant NPSTs by using bioinformatic tools are easy. However, when the required sequence alignment markers are experimentally determined, their functions have been proven to be reliable indicators of biosynthetic pathways and compound production [81]. eSNaPD has NPST analysis features that can organize the data to enable archiving and comparative study of NPST datasets from a number of environments. Such approaches for investigating biosynthetic complexity in environmental and metagenomic library collections are low-cost and computationally intensive. However, only a small number of PCR amplicons are sequenced and then computed instead of entire genomes.

(ii) In a random sequence (shotgun sequencing), larger fragments of DNA are broken into shorter fragments and randomly sequenced for multiple rounds (Figure 1). After separation, the entire sequenced are re-assembled using several overlapping of the sequenced fragments [85]. Necessary steps required for the shotgun sequencing are (i) high-quality metagenomic DNA isolation, (ii) random metagenomic DNA fragmentation (ultra-sonication), (iii) size fractionation (electrophoresis), (iv) metagenomic library construction, (v) paired-end sequencing, and (vi) assembly in silico of sequenced segments. Random metagenomic sequence is preferred for characterizing the genome content of bacteria, archaea, and viruses and their secondary metabolic pathways [86]. This method allows complete genomic DNA to be recovered, possibly from sources without in vitro cultivation. Random sequencing has benefits over other methods in that it speeds up the gene mapping and takes less effort to map. Conversely, it mechanistically lacks positive sorting and recognition of the biological functionaries’ active classes. Furthermore, certain drawbacks are often related when applied to metagenomes composed of multiple repeated DNA sequences, and thus provide inexact evidence for the genome sequenced [87].

With the cost of DNA synthesis falling, synthetic biology will substitute the desire to isolate cells from the source environments, in order to have access to heterologous expression of BGCs. Calcium-dependent antibiotic malacidins active against multidrug-resistant pathogens are commonly encrypted in soil microbiomes and discovered via a sequencing-based metagenomic strategy [88,89].

However, additional quality controls are needed because the risk of chimeras in metagenomic sequence clusters is always present, particularly with fast-evolving and repeated genes in BGCs that encode modular polyketide synthases, for example. Particularly, metagenomics is a transient research area in its present ‘short read–based’ nature, since the technological constraints that currently characterize it are likely to be overcome quickly. Despite the expected new technologies in the coming years, the shotgun sequencing method for metagenomic BGC classification may completely substitute tag sequencing [90].

Nevertheless, many BGCs surpass the normal insert size of the cosmid, and thus many overlapping DNA cosmid clones covering the entire BGCs must be rearranged into a bacterial artificial chromosome (BAC) (stitching) by the commonly employed transformation-associated recombination process (TAR) or Red/ET-mediated recombineering [91]. Subsequently, the reassembled BGC may be moved from the yeast/*E. coli* to separate bacterial hosts for heterologous expression and further pathway optimization (Figure 1). As a result, this technique is more a heterologous language dependent on a metagenomic library than expression of metagenomic samples.

## 4. Single-Cell Metagenomics

As opposed to sequencing-based metagenomics, single-cell metagenomics analyzes the genomes of individual cells retrieved from the environmental samples, which examines the overall population DNA with subsequent bioinformatics-based binning of obtained sequences among individual microbial organisms [26,27,28]. One significant benefit of this strategy is that it is new and effective for analyzing uncultured microorganisms in environmental samples and determining whether metabolites as well as other genes are linked with phylogenetic knowledge, which has been challenging for general metagenomic techniques. Thus, it is considered as a powerful complement to the general metagenomics techniques (Figure 1).

### 4.1. Cell Sorting and Single-Cell Sequencing

Isolation of single cells from environmental samples is the first step of single-cell metagenomics, generally depending on flow cytometry, microfluidic instruments, and micromanipulation [92].

PCR can then be applied specifically on the cells in order to connect the feature with taxonomy (digital PCR). For example, after screening cells with a multiplex microfluidic system, genes from primary metabolism in a termite gut culture were linked in this way with a spirochete symbiont [93].

In some cases, it is indeed helpful to amplify a cell’s genome before additional study, a technique known as whole-genome amplification (WGA). It is possible due to bacteriophage polymerase, which produces microgram quantities of DNA in only one reaction, linking multiple displacement amplification (MDA) [94]. This method revealed the origins of an uncommon community of PKSs found in microbial–sponge interactions and supposed to play a role in the biosynthesis of methyl-branched fatty acids [95,96]. Single cells were processed in such experiments by dissociating sponge tissue and filtering bacteria into 96-well plates using flow cytometry [97,98].

It was also noted that some WGA approaches set up in human genetic study could provide methodological directions for future single-cell genome sequencing. For instance, MALBAC (Multiple Annealing and Looping Based Amplification Cycles) was exemplified for single-cell genomic sequencing [99,100]. It required specifically-designed primers having a 27-nt common sequence, followed by eight random nucleotides. Using these primers for PCR amplification of the single cell genome, the resultant amplicons could form loops due to the presence of the two complementary terminals, which may be resistant to subsequent amplification and hybridization. Combining next-generation sequencing, this approach was expected to be applied into uncultured microorganisms.

### 4.2. Screening of NP Gene Clusters in Single-Cell Metagenomics

Single-cell metagenomics as an important tool for studying environmental bacteria’s lifestyles has a lot of potentials in NP analysis [100,101,102]. The availability of single-cell genome sequences provides rich genetic resources for NP exploration from uncultured microorganisms. For example, study of uncultured sponge-symbiotic *Entotheonella* single-bacterial genomics confirms a rich resource of different specialized metabolites in sponges [96].

Marine sponges were considered to have plenty of microbes with giant potentials to produce NPs. Following MDA, DNA samples were screened that revealed the PKSs belong to members of the rare candidate phylum “*Poribacteria*” [103], which is extensively connected with sponges but almost missing in other habitats [97]. The expression of the PKS genes was confirmed by partial sequencing of the genome of single cell amplified poribacteria [98]. This was the earliest research to use single-cell metagenomics to chart the positions of NP genes. Although the construction of chimeric amplicons during amplification may render genomic assembly difficult, there have been reports of entire genomes being assembled from single cells [104].

The drawbacks of this method are related to single cell isolation and lysis, contamination susceptibility, and imperfect and unevenly amplifying single-cell genomes (single amplified genomes, SAG) [18]. As a consequence, sequenced SAGs are commonly characterized by a series of contigs that only occupy a small portion of the whole genome [26]. The combination of data from a sequence of several SAGs from cells of a single species might overcome such challenges [13]. Rinke et al. sequenced 201 SAGs comprising 29 uncultured lineages of “microbial dark matter” as an example of active implementation of single-cell metagenomics [105,106].

Using single cell genome sequences containing NP BGCs, the combining of DNA synthesis and BGC assembly with heterologous expression will be applied to establish biosynthetic pathways of target NPs hidden in uncultured microorganisms [90].

Both single-cell metagenomics and sequencing-based metagenomics yield DNA sequences from uncultured microorganisms, but which are radically distinct. SAGs from single-cell metagenomics are genomes of individual organisms, while sequencing-based metagenomics bins are aggregate genomes of a genetically heterogeneous community. The latter is ineffective since it does not allow for the full genetic capacity of various strains of the same species present in a community [18].

## 5. Conclusions

Different environmental habitats have different ecosystems and biodiversity, and have different potentials for NP discovery. For example, in animal guts, many NPs were accumulated by gut microorganisms related to many physiological processes of animals [107], and metagenomic analysis revealed a high diversity in animal gut microbiome [108]. Strategies for multi-dimensional visualization of metagenomic data have been developed [109]. An Influential international project (The Earth Microbiome Project) launched in August 2010 has been attempting to classify worldwide microbial diversity in terms of phylogeny and function [110].

Though the number of NPs isolated from environmental samples is still extremely low, a high number of NP BGCs have been identified from uncultivated microorganisms that will still be great treasure troves for NP discovery in this field [111,112]. Among the approaches used in study on environmental samples or uncultured microorganisms, metagenomics provides innovative tools for discovering natural resources that has the ability to access historically undiscovered biosynthetic diversity. Latest advancements in bioinformatics techniques and sequencing technologies (e.g., pyrosequencing, nanopore sequencing) have enabled the analysis of whole microbial populations made up of unknown organisms by metagenomics.

In the coming years, there will be a combination of fast-developing long-read technologies and new bioinformatics methods that will enable scientists to acquire complete genome sequences of all or complete gene clusters of NPs, but the least influential members of the microbial population by scanning them directly from the broader metagenomic data collection. In the near future, hybrid methods that incorporate assembly and tag sequences may also be of big benefit; in such a hybrid technique, integrated metagenomic contigs may be used to classify new candidate groups of BGCs for which primers are then built for further investigation through tag sequences, phylogenomic analysis, and cosmid sequences.

Thus, metagenomics is opening up unique opportunities for the identification of bioactive compounds and several valuable compounds are anticipated to be identified and exploited in the near future. Interdisciplinary studies into bioactive compounds synthesized by NRPS and PKS will continue to be broadening in order to access the wide range of secondary metabolites obtained by microorganisms (including uncultured or yet-uncultured species). Moreover, metatranscriptomics, metaproteomics, and metabolomics techniques may help researchers better understand how microbes respond in a community [113].

Even by only changing the cultivation conditions or establishing novel screening methods, uncultured microorganisms could become culturable and accessible, and this is still proven to be reliable for the discovery of some new NPs, exampled by the exploration of teixobactin, considered as a new generation of antibiotic against multi-drug resistant (MDR) pathogens [114]. Single-cell genome mining of NPs for insight into environmental samples or uncultured microorganisms was exemplified with the biosynthesis of the extraordinary complex hypermodified polytheonamide-type compounds, which BGC sequence was identified originally from a sponge-symbiotic uncultivated species in *Entotheonella.* The availability of such a BGC sequence guided the discovery of similar biosynthetic pathways in some culturable bacteria, suggesting that some BGCs are subject to extensive horizontal gene transfer between culturable and uncultivated sources [115].

With advances in the synthetic biological methods used in DNA synthesis, DNA assembly, and direct cloning, and refactoring of large BGCs, combining of updated bioinformatics, combinatorial chemistry, and synthetic biological methods with metagenomic approaches would further promote the development or man-made modification of potentially necessary medicinal compounds from uncultured microorganisms.

## Figures and Tables

**Figure 1 molecules-26-02977-f001:**
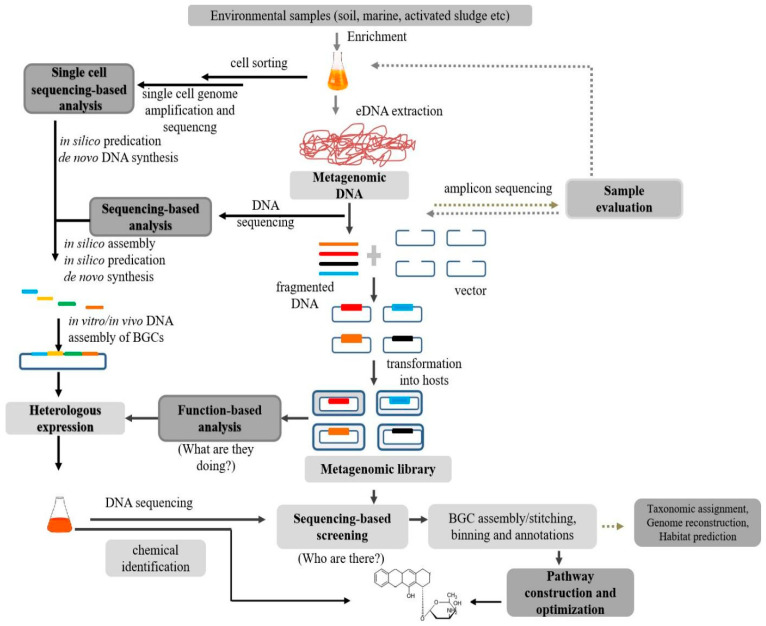
Graphical representation of approaches for NP discovery from environmental samples. Enrichment and evaluation of environmental samples are necessary steps prior to metagenomics analysis. Function-based metagenomics analysis includes library construction, heterologous expression, and function-based screening. Sequencing-based metagenomics analysis requires eDNA sequencing, DNA synthesis and assembly, and heterologous expression of target clones. Single-cell metagenomics analysis includes cell sorting, single cell genomic DNA amplification, genome sequencing, and DNA synthesis and assembly of NP biosynthetic gene clusters (BGGs) of interest, then expression and chemical identification.

**Figure 2 molecules-26-02977-f002:**
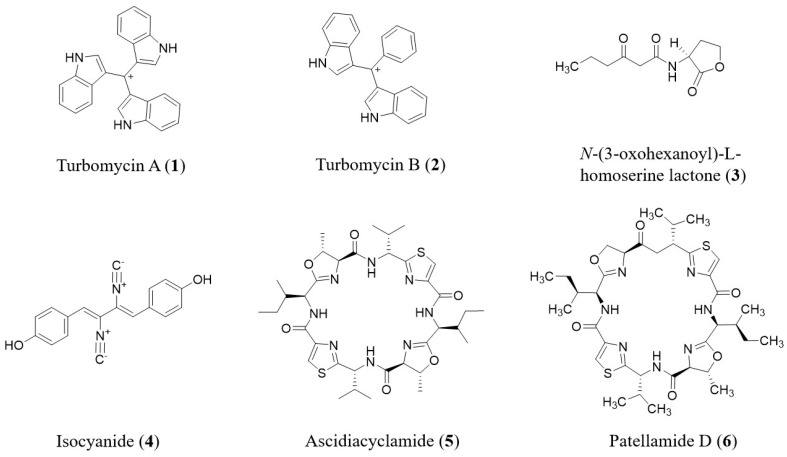
Natural products discovered by function-based screening of metagenomic-library clones.

**Figure 3 molecules-26-02977-f003:**
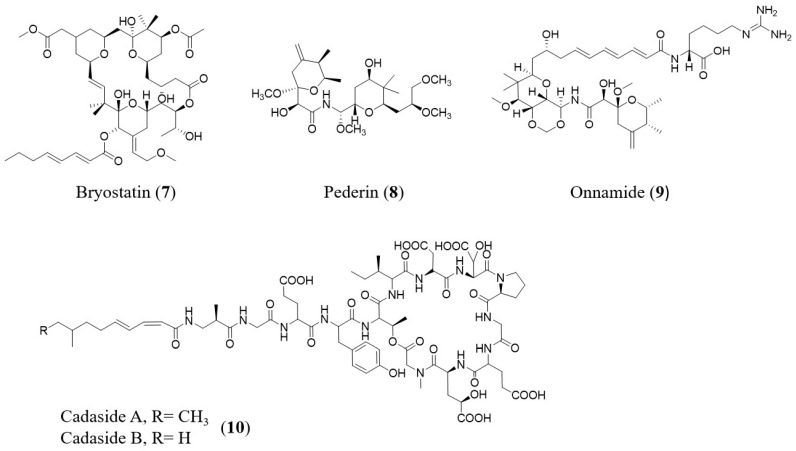
Natural products identified/studied by sequencing-based approaches.

**Table 1 molecules-26-02977-t001:** Antibiotics discovered by function-based screening of metagenomic clones.

Compound	Habitat	Library Type	Reference
Fasamycin A and B	Soil	Cosmid	[40]
Indirubin	Soil	Fosmid	[41]
Terragine	Soil	Cosmid	[42]
Turbomycins A and B	Soil	BAC	[43]
Violacein	Soil	Cosmid	[40]

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
