# Peer review of "Strategies for Natural Products Discovery from Uncultured Microorganisms"

_molecules, 2021, doi:10.3390/molecules26102977_

Round 1

Reviewer 1 Report

I think that the originality of the manuscript is medium. A couple of similar reviews has been published. However, some researchers in this field may be interested to take a look on. 

Minor comments

-Page 2 Lines 45 and 46. It should be clarified that the number of cells is estimated.

-The name of genes must be written in italics. For example see page 2. line 77. Make proper corrections along the manuscript.

-Page 5. Line 177. The name of the specie must be write in full at the beginning of a sentence. Escherichia coli.

Reviewer 2 Report

The topic of the review is a relevant area of research in the field of ecology, microbiology, biochemistry, etc. However, the presented review does not make significant contribution to the field. Metagenomics, metabolomics, and sequencing are methods that have already become standard for the search for uncultivated microorganisms and their functional genes, including for the purpose of discovering new natural products. Many reviews and experimental articles are devoted to this actively developing topic. However, among 134 references in the manuscript, only 11 are dated 2018–2020. Moreover, in the introduction, the authors cite a recent review publication similar in topic to the presented review (Kalkreuter et al., 2020). In addition, the article contains a number of misprints and significant formatting mistakes. For example:

  1. Throughout the text, the authors introduce some abbreviations (NP, BGC), but later they do not use them (lines 34, 140, 224, 371, 375, etc.). In addition, the authors introduce abbreviations for phrases that are no longer repeated in the text (lines 448, 506). Due to the abundance of abbreviations, it is recommended to make a section “Abbreviations”.
  2. The authors use both “uncultured microorganisms” and “uncultivated microorganisms”. They are synonyms, and it is worth using one of them throughout the entire text.
  3. What is the difference between “functional screening” and “function-based screening”? If they are synonyms, then only one of them should be used.
  4. What is the difference between “sequencing-based” and “sequence-based”? If they are synonyms, then only one of them should be used.
  5. Table 1 is devoted to the antibiotic properties of the compounds, why is the “Function” column needed?
  6. In Table 2, some words are in bold, why?
  7. All tables have a “Serial” column, what is it and why is it needed?
  8. There are some mistakes in references to figures (lines 277, 330). If authors want to refer to the formula of a certain compound in a figure, the reference should be made as follows “(1, Figure 1)”, where “1” is the number of the compound in the figure.
  9. In Figure 3, the caption for the cadaside A is incorrect.
  10. Figures should be placed in the text near to the first time they are cited.
  11. There are some mistakes in the writing of Latin names and words (lines 161, 195-196, 251, 263, 456).
  12. Why is the text repeated in “Funding” and “Acknowledgments”? It should to be divided into the sections according to the meaning.
  13. The bibliography does not correspond to the requirements of the journal.
  14. There are a lot of misprints throughout the text (lines 23, 77, 219, 239, 287, 295, 330).

Therefore, the article would not to be recommended for publication in Molecules.

Reviewer 3 Report

I enjoyed reading the review of metagenomic methods for natural products discovery by Alam, et al.  Their topic is pertinent and should be of interest to people who wish to keep current with the natural products space.  There were several cases where the organization of the manuscript could be improved (see below), and the wording and grammar were confusing at time.  I have the following suggestions to improve the manuscript:

  1. The second paragraph of section 1.2 seems a bit redundant with 1.1.
  2. Please mention the expression of heterologous sigma factors for improving expression of eDNA (https://www.nature.com/articles/ncomms8045)
  3. Please briefly mention the throughput of different screening systems (e.g. visual inspection, growth-based assays, FACS, HPLC, etc).
  4. Casaside A in Fig. 3 seems distorted.
  5. The last paragraph in section 4 is out of place.
  6. The second to last paragraph in section 4 seems more forward-looking. Perhaps it belongs in the conclusion?
  7. The document needs some editing for correct English grammar and flow.

Round 2

Reviewer 2 Report

Despite the fact that the authors have worked on the content and design of the article, there are still misprints and significant formatting mistakes, especially in References:

  1. Some of the titles of chapters and subchapters are with a dot at the end; some of them are without it. In addition, the formatting of the subchapter 2.2.2 is different. Please, clarify formatting of the titles in the Journal.
  2. There are misprints (lines 136, 387, 339)
  3. The BCG abbreviation should be in line 321.
  4. Table 2 shows examples of natural products found using sequencing-based approaches. However, 3 of them were obtained from cultured microorganisms. The use of these examples does not seem logical, because the article is devoted to uncultivated microorganisms.
  5. The generic name of microorganisms should to be abbreviated on subsequent mentions (line 165).
  6. The names of genes should to be written in italics (lines 175, 247)
  7. There are cadasides A and B in the text, but there is only cadaside A in the figure (line 322).
  8. Caption of the Figure 3 is in smaller font.
  9. In line 388 it is not clear for what subject there is “is”.
  10. The authors do not provide a transcript for the DNA abbreviation, but provide it for PCR. Why? The abbreviation PCR is ubiquitous, like DNA.
  11. References in the text do not correspond to the list of references (lines 86, 183, 215, 244, 263, 463).
  12. In the References, some references are repeated 2 times (9 and 11, 70 and 81).
  13. Many references do not have pages and doi.
  14. The reference 42 is not formatted correctly.
  15. The references 66, 95, and 106 have misprints.
  16. For what purpose the authors provide the reference 111? The text is about analysis of microbial populations using bioinformatics and sequencing technologies; but the reference article is about detection of SARS-CoV-2 viruses (line 490). The authors should to pay more attention when use some articles.

I still believe that the contribution of this article to the research field is low. However, the article, subject to significant revision, perhaps, could become an interesting part of the special issue "Bioactive Natural Products from Microorganisms".
